# Mortality causes universal changes in microbial community composition

Clare I. Abreu[1], Jonathan Friedman [2], Vilhelm L. Andersen Woltz[1] & Jeff Gore [1]

All organisms are sensitive to the abiotic environment, and a deteriorating environment can cause extinction. However, survival in a multispecies community depends upon interactions, and some species may even be favored by a harsh environment that impairs others, leading to potentially surprising community transitions as environments deteriorate. Here we combine theory and laboratory microcosms to predict how simple microbial communities will change under added mortality, controlled by varying dilution. We find that in a two-species coculture, increasing mortality favors the faster grower, confirming a theoretical prediction. Furthermore, if the slower grower dominates under low mortality, the outcome can reverse as mortality increases. We find that this tradeoff between growth and competitive ability is prevalent at low dilution, causing outcomes to shift dramatically as dilution increases, and that these two-species shifts propagate to simple multispecies communities. Our results argue that a bottom-up approach can provide insight into how communities change under stress.

[1] Department of Physics, Massachusetts Institute of Technology, Cambridge 02139 MA, USA. [2] Department of Plant Pathology and Microbiology, The Hebrew University of Jerusalem, Rehovot 7610001, Israel. Correspondence and requests for materials should be addressed to J.G. (email: gore@mit.edu)

Ecological communities are defined by their structure, which includes species composition, diversity, and interactions[1]. All such properties are sensitive to the abiotic environment, which influences both the growth of individual species and the interactions between them. The structure of multispecies communities can thus vary in complex ways across environmental gradients[2–7]. A major challenge is therefore to predict how a changing environment affects competition outcomes and alters community structure. In particular, environmental deterioration can radically change community structure. Instances of such deterioration include antibiotic use on gut microbiota[8], ocean warming in reef communities[9], overfishing in marine ecosystems[10], and habitat loss in human-modified landscapes[11]. Such disturbances can affect community structure in several ways, such as allowing for the spread of invasive species[12], causing biodiversity loss and mass extinction[13,14], or altering the interactions between the remaining community members[15,16]. For example, a stable ecosystem can be greatly disrupted by the removal of a single keystone species, potentially affecting species with which it does not directly interact[17–19].

A common form of environmental deterioration is increased mortality, which can be implemented in the laboratory in a simple way. In fact, the standard method of cultivating and coculturing bacteria involves periodic dilution into fresh media, a process that necessarily discards cells from the population. The magnitude of the dilution factor determines the fraction of cells discarded and therefore the added mortality rate, making environmental harshness easy to tune experimentally.

The choice of dilution factor often receives little attention, yet theoretical models predict that an increased mortality rate experienced equally by all species in the community can have dramatic effects on community composition. In particular, it is predicted that such a global mortality rate will favor the faster-growing species in pairwise coculture, potentially reversing outcomes from dominance of the slow grower to dominance of the fast grower[1,20,21] as mortality increases. Indeed, there is some experimental support for such reversals in chemostat experiments with microbial species with different growth rates[22–24]. A less-explored prediction is that if a high mortality rate causes a competitive reversal, the coculture will also result in either coexistence or bistability (where the winner depends on the starting fraction) at some range of intermediate mortality[25–27]. Missing from the literature is a systematic study that probes both of these predictions with an array of pairwise coculture experiments across a range of dilution rates. In addition, little is known about how mortality will alter the composition of multispecies communities.

In this paper, we report experimental results that expand upon the prior literature regarding the effect of dilution on pairwise outcomes, and we use the pairwise outcomes to develop a predictive understanding of how multispecies community composition changes with increased dilution. First, pairwise coculture experiments with five bacterial species confirmed that (1) increased mortality favors the fast grower and can reverse the winner, or the only remaining species at the end of the experiment, from slow grower to fast grower, and (2) at intermediate dilution rates, either coexistence or bistability occurs, where from many starting fractions, the two species' abundances either converge to a stable fraction or diverge to either species winning, respectively. We measure species' growth rates by growing cells from low density in monoculture; fast growers reach a threshold density more quickly than slow growers. We define the competitive ability of a species as its average fraction after being cocultured in pairs with each of the other species for multiple dilution cycles. Interestingly, we find that a pervasive tradeoff between growth rate and competitive ability in our system favors

slow growers in high-density, low-dilution environments, leading to striking changes in outcomes as mortality increases. Second, to bridge the pairwise results to three- and four-species communities, we employ simple predictive pairwise assembly rules[28], where we find that the pairwise outcomes such as coexistence and bistability propagate up to the multispecies communities. Our results highlight that the seemingly complicated states a community adopts across a mortality gradient can be traced back to a predictable pattern in the outcomes of its constituent pairs.

## Results

**Three-species community exhibits wide range of stable states.** To probe how a changing environment affects community composition, we employed an experimentally tractable system of soil bacteria coculture experiments subject to daily growth/dilution cycles across six dilution factors (Fig. 1a). We selected five species of soil bacteria: *Enterobacter aerogenes* (*Ea*), *Pseudomonas aurantiaca* (*Pa*), *Pseudomonas citronellolis* (*Pci*), *Pseudomonas putida* (*Pp*), and *Pseudomonas veronii* (*Pv*) (Supplementary Fig. 10). These species have been used in previous experiments by our group, which did not vary dilution factor[28,29]. All five species grow well in our defined media containing glucose as the primary carbon source (see "Methods") and have distinct colony morphology that allows for measuring species abundance by plating and colony counting on agar.

We began by competing three of the five species, *Ea*, *Pci*, and *Pv*, for seven 24-h cycles under six different dilution factor regimes. To assay for alternative stable states, each dilution factor condition was initialized by four different starting fractions (equal abundance as well as prevalence of one species in a 90–5–5% split). Despite the simplicity of the community and the experimental perturbation, we observed five qualitatively different outcomes corresponding to different combinations of the species surviving at equilibrium (Fig. 1b). At the highest and lowest dilution factors, one species excludes the others at all starting fractions (*Pv* at low dilution, *Ea* at high dilution). Two coexisting states (*Ea*–*Pv* and *Ea*–*Pci*) occur at medium low ($10^2$) and medium high ($10^4$) dilution factors, again independent of the starting fractions of the species. However, at intermediate dilution factor ($10^3$), we found that the surviving species depended upon the initial abundances of the species. At this experimental condition, the system displays bistability between the two different coexisting states (*Ea*–*Pv* and *Ea*–*Pci*) that were present at neighboring dilution factors. These three species therefore display a surprisingly wide range of community compositions as the mortality rate is varied.

**Two-species model predicts that mortality favors faster grower.** To make sense of these transitions in community composition, we decided to first focus on two-species competitions, not only because they should be simpler but also because prior work from our group gives reason to believe that pairwise outcomes are sufficient for predicting multispecies states[28]. Accordingly, we used a simple two-species Lotka–Volterra (LV) competition model with an added mortality term $\delta N_i$ experienced equally by both species[21]:

$$\dot{N}_i = r_i N_i \left(1 - N_i - \alpha_{ij} N_j\right) - \delta N_i \qquad (1)$$

where $N_i$ is the density of species $i$ (normalized to its carrying capacity), $r_i$ is the maximum growth rate of species $i$, and the competition coefficient $\alpha_{ij}$ is a dimensionless constant reflecting how strongly species $i$ is inhibited by species $j$ (Fig. 2). This model can be re-parameterized into the LV model with no added mortality, where the new competition coefficients $\tilde{\alpha}_{ij}$ now depend

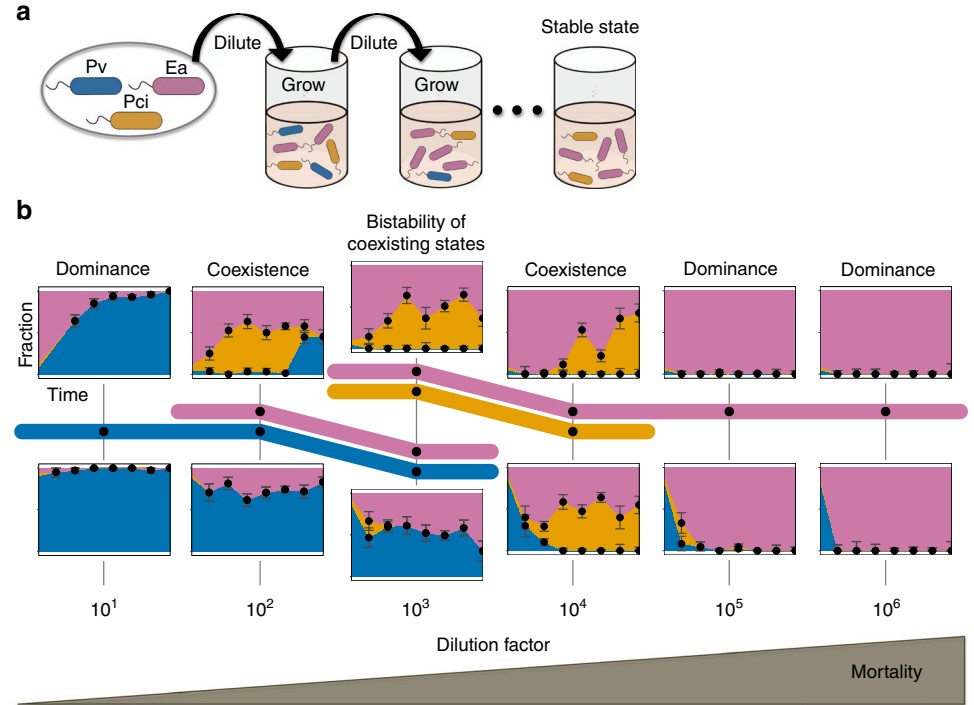

**Fig. 1** Increasing dilution causes striking shifts in a three-species community. **a** To probe how added mortality changes community composition, we cocultured three soil bacteria over a range of dilution factors. Cells were inoculated and allowed to grow for 24 h before being diluted into fresh media. This process was continued for 7 days, until a stable equilibrium was reached. The magnitude of the dilution factor ($10$–$10^6$) determines the fraction of cells discarded, and thus the amount of added mortality. **b** We began with a three-species community (*Enterobacter aerogenes* (*Ea*), *Pseudomonas citronellolis* (*Pci*), and *Pseudomonas veronii* (*Pv*)), initialized from four starting fractions at each dilution factor. The outcomes of two of the starting fractions are shown (see Supplementary Fig. 8b for remaining starting fractions), along with a subway map, where survival of species is represented with colors assigned to each species. Black dots indicate where data were collected, while colors indicate the range over which a given species is inferred to survive. Species *Pv* dominates at the lowest dilution factor, and *Ea* dominates at the highest dilution factors. The grouping of two colors represents coexistence of two species, whereas the two levels at dilution factor $10^3$ indicate bistability, where both coexisting states, *Ea*–*Pv* and *Ea*–*Pci*, are stable and the starting fraction determines which stable state the community reaches. Error bars are the SD of the beta distribution with Bayes' prior probability (see "Methods"). Source data are provided as a Source Data file

upon $r_i$ and $\delta$ (Supplementary Note 1, Supplementary Fig. 11):

$$\dot{\tilde{N}} = \tilde{r}_i \tilde{N}_i \left( 1 - \tilde{N}_i - \tilde{\alpha}_{ij} \tilde{N}_j \right) \qquad (2)$$

$$\tilde{\alpha}_{ij} = \alpha_{ij} \frac{\left( 1 - \frac{\delta}{r_j} \right)}{\left( 1 - \frac{\delta}{r_i} \right)} \qquad (3)$$

The outcome of competition—dominance, coexistence, or bistability—simply depends upon whether each of the $\tilde{\alpha}$ are greater or less than one, as in the basic LV competition model[21]. Stable coexistence occurs when both $\tilde{\alpha}$ coefficients are less than one, bistability when both are greater than one, and dominance/exclusion when only one coefficient is greater than one.

In this model, it is possible for a slow grower ($N_s$) to outcompete a fast grower ($N_f$) if the slow grower is a strong competitor ($\alpha_{fs} > 1$) and the fast grower is a weak competitor ($\alpha_{sf} < 1$) (Fig. 2). However, the competition coefficients change with increasing mortality $\delta$ in a way that favors the fast grower: $\tilde{\alpha}_{fs}$ shrinks and $\tilde{\alpha}_{sf}$ grows, eventually leading the fast grower to outcompete the slow grower. A powerful way to visualize this change is to plot the outcomes as determined by the competition coefficients (Fig. 2c); increasing mortality causes the outcome to traverse a 45° trajectory through the phase space, leading to the fast grower winning at high mortality. At intermediate mortality, the model predicts that the two species will either coexist or be bistable. This model therefore makes very clear predictions regarding how pairwise competition will change under increased mortality, given the aforementioned slow grower advantage at low mortality.

**Dilution experiments confirm predictions about mortality**. To test these predictions in the laboratory, we performed all pairwise coculture experiments at multiple dilution factors and starting fractions of our five bacterial species: *Pp*, *Ea*, *Pci*, *Pa*, *Pv* (listed in order from fastest- to slowest-growing species). We find that these pairwise outcomes change as expected from the LV model, where increased dilution favors the fast grower (Supplementary Fig. 1). For example, in *Ea*–*Pv* competition we find that *Pv*, despite being the slower grower, is able to exclude *Ea* at low dilution rates (Fig. 3b, left panel). From the standpoint of the LV model, *Pv* is a strong competitor despite being a slow grower in this environment. However, as predicted by the model, at high dilution rates the slow-growing *Pv* is excluded by the fast-growing *Ea* (Fig. 3b, right panel). Importantly, *Pv* is competitively excluded at a dilution factor of $10^4$, an experimental condition at which it could have survived in the absence of a competitor. Finally, and again consistent with the model, at intermediate dilution rates we find that the *Ea*–*Pv* pair crosses a region of coexistence, where the two species reach a stable fraction over time that is not a function of the starting fraction (Fig. 3b, middle panel). The *Ea*–*Pv* pair therefore displays the transitions through the LV phase space in the order predicted by our model (Fig. 3a–d).

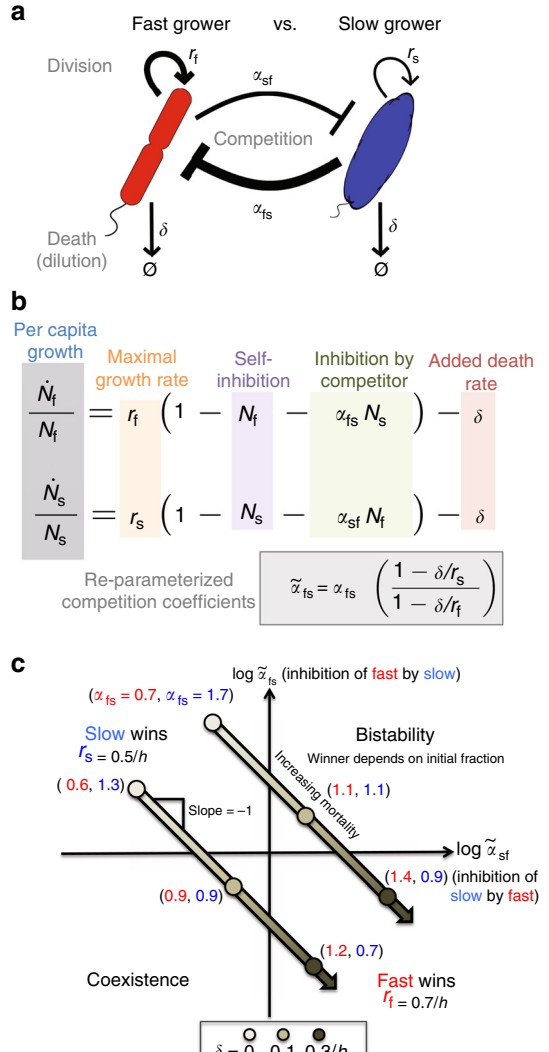

**Fig. 2** An increasing global mortality rate is predicted to favor the fast grower. **a**, **b** Here we illustrate the parameters of the Lotka–Volterra (LV) interspecific competition model with added mortality: population density $N$, growth $r$, death $\delta$, and the strengths of inhibition $\alpha_{sf}$ and $\alpha_{fs}$ (subscript f for fast grower and s for slow grower). Here we assume a continuous death rate, but in the model, the outcome is the same for a discrete process, such as our daily dilution factor (Supplementary Note 2). The width of arrows in **a** corresponds to an interesting case that we observe experimentally, in which the fast grower is a relatively weak competitor. **c** The outcomes of the LV model without mortality depend solely upon the competition coefficients $\alpha$, and the phase space is divided into one quadrant per outcome. If the slow grower is a strong competitor, it can exclude the fast grower. Imposing a uniform mortality rate $\delta$ on the system, however, favors the faster grower by making the re-parameterized competition coefficients $\tilde{\alpha}$ depend on $r$ and $\delta$. Given that a slow grower dominates at low or no added death, the model predicts that coexistence or bistability will occur at intermediate added death rates before the outcome transitions to dominance of the fast grower at high added death (Supplementary Note 1). Two numerical examples show that the values of $\alpha$ (in the absence of added mortality) determine whether the trajectory crosses the bistability or coexistence region as mortality increases

The LV model predicts that other pairs will cross a region of bistability rather than coexistence, and indeed this is what we observe experimentally with the *Pci–Pv* pair (Fig. 3e–h). Once again, the slow-growing *Pv* dominates at low dilution factor yet is

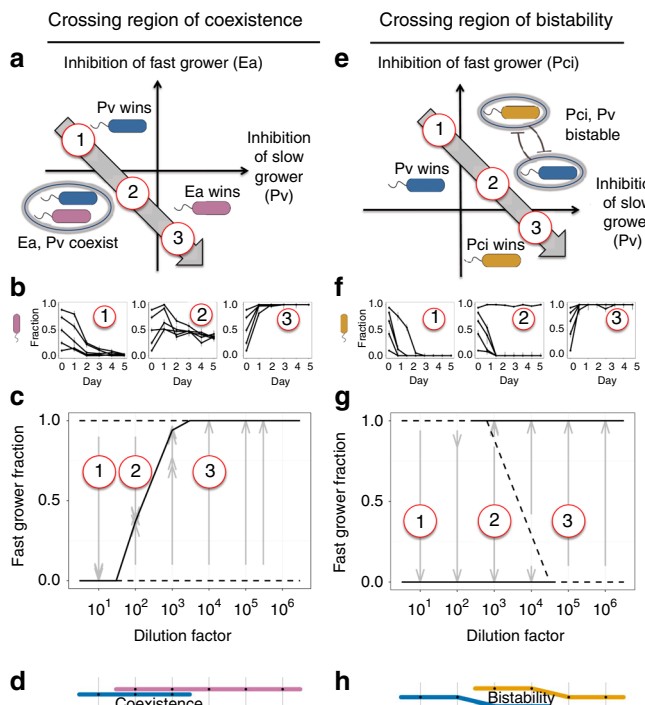

**Fig. 3** In pairwise coculture experiments, increasing dilution favors the faster grower. **a** Experimental results are shown from a coculture experiment with *Pv* (blue) and *Ea* (pink). **b** Left panel: Despite its slow growth rate, *Pv* excludes faster grower *Ea* at the lowest dilution factor. Middle panel: Increasing death rate causes the outcomes to traverse the coexistence region of the phase space. Right panel: As predicted, fast-growing *Ea* dominates at high dilution factor. Error bars are the SD of the beta distribution with Bayes' prior probability (see "Methods"). **c** An experimental bifurcation diagram shows stable points with a solid line and unstable points with a dashed line. The stable fraction of coexistence shifts in favor of the fast grower as dilution increases. Gray arrows show experimentally measured time trajectories, beginning at the starting fraction and ending at the final fraction. **d** A "subway map" denotes survival/extinction of a species at a particular dilution factor with presence/absence of the species color. **e**, **f** *Pv* outcompeted another fast grower *Pci* (yellow) at low dilution factors, but the pair became bistable instead of coexisting as dilution increased; the unstable fraction can be seen to shift in favor of the fast grower (**g**). **h** Two levels in the subway map show bistability. Source data are provided as a Source Data file

excluded at high dilution factor. However, at intermediate dilution factors this pair crosses a region of bistability, in which the final outcome depends upon the starting fractions of the species. The LV model with added mortality therefore provides powerful insight into how real microbial species compete, despite the many complexities of the growth and interaction that are necessarily neglected in a simple phenomenological model.

Indeed, a closer examination of the trajectory through the LV phase space of the *Pci–Pv* pair reveals a violation of the simple outcomes allowed within the LV model. In particular, at dilution factor $10^2$ we find that when competition is initiated from high initial fractions of *Pci* that *Pv* persists at low fraction over time (Fig. 3g). This outcome, a bistability of coexistence and exclusion (rather than of exclusion and exclusion), is not an allowed outcome within the LV model (modifications to the LV model can give rise to it, as shown by ref. [30]). This subtlety highlights that the transitions (e.g., bifurcation diagrams in Fig. 3c, g) can be more complex than what occurs in the LV model but that nonetheless the transitions within the LV model represent a baseline to which quantitative experiments can be compared.

**Tradeoff between growth rate and competitive ability observed.**
The model predicts that mortality will reverse coculture outcomes if and only if a slow grower excludes a fast grower at low or no added death, exhibiting a tradeoff between growth and competitive ability. Changes in outcome are therefore most dramatic when a strongly competing slow grower causes the trajectory to begin in the upper left quadrant of the phase space (Fig. 3a, e), allowing it to move through other quadrants as mortality increases. Indeed, in the pairwise experiments described above, the slowest-growing species, *Pv*, is a strong competitor at low dilution factor. To probe this potential tradeoff more extensively, we compared the growth rates of our five species in monoculture (Supplementary Figs. 3, 4, and 5) to their competitive performance at low dilution factor. In seven of the ten pairs, the slower grower excluded the faster grower, and the other three pairs coexisted (Supplementary Fig. 1). We therefore find that our five species display a pervasive tradeoff between growth rate and competitive ability, possibly because the slower-growing species fare better in high-density environments that reach saturation.

To visualize how competitive success changes with dilution factor, we defined the competitive score of each species to be its mean fraction after reaching equilibrium in all pairs in which it competed. The aforementioned tradeoff can be seen as an inverse relationship between growth rate and competitive score at the lowest dilution factor (Fig. 4a). As predicted, the performance of the fast-growing species increases monotonically with increasing dilution factors (Fig. 4b). Competitive superiority of the slowest grower (*Pv*) at low dilution rates transitions to the next slowest (*Pa*) at intermediate rates, before giving rise to dominance of the fastest growers (*Pci*, *Ea*, *Pp*) at maximum rates (Fig. 4b–d). We therefore find that the mortality rate largely determines the

importance of a species' growth rate to competitive performance in coculture experiments.

**Pairwise outcomes predict multispecies states.** Now that we have an understanding of how pairwise outcomes shift in response to increased mortality, we return to the seemingly complicated set of outcomes observed in our original three-species community (Fig. 1). In a previous study[28], we developed community assembly rules that allow for prediction of species survival in multispecies communities from the corresponding pairwise outcomes. These rules state that in a multispecies coculture, a species will survive if and only if it coexists with all other surviving species in pairwise coculture. If one or more bistable pairs are involved in a multispecies community, the assembly rules allow for either of the stable states. We see that the seemingly complicated trio outcomes follow from these assembly rules applied to our corresponding pairwise outcomes at all dilution factors (Fig. 5). For example, at the lowest dilution factor (10), *Ea–Pci* coexist, but each of these species is excluded by *Pv* in pairwise coculture, thus leading to the (accurate) prediction that only *Pv* will survive in the three-species coculture experiment. In addition, we observe that the bistability of *Pci–Pv* at dilution factor $10^3$ propagates up to lead to bistability in the trio but with each stable state corresponding to coexistence of two species. The only trio outcome not successfully predicted by the rules is the extinction of *Pci* at a dilution factor of $10^5$ (Fig. 5d, Supplementary Fig. 8). Our analysis of pairwise shifts under increased mortality therefore provides a predictive understanding of the complex shifts observed within a simple three-species bacterial community.

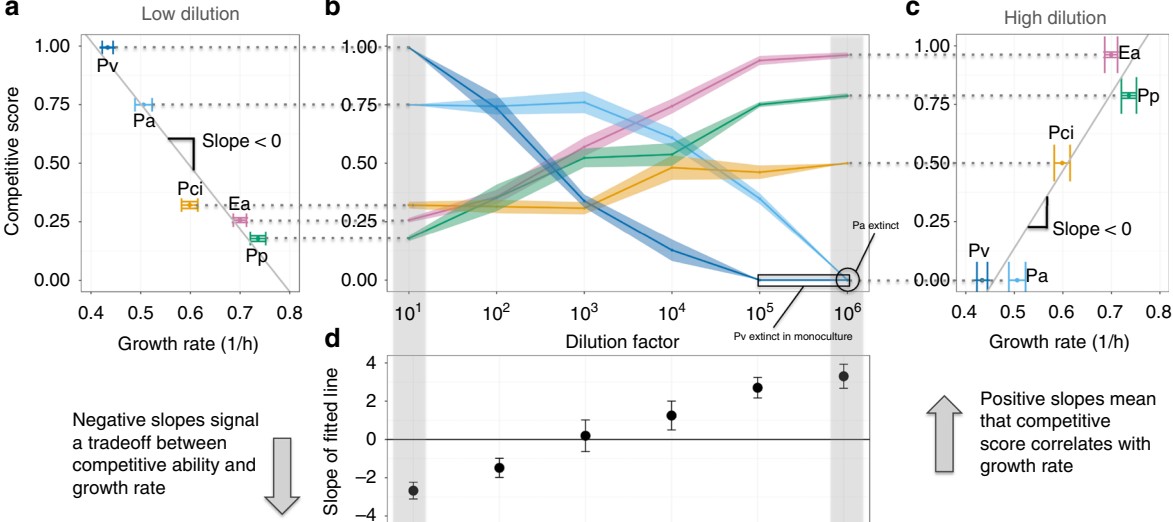

**Fig. 4** Tradeoff between growth and competitive ability leads to dependence of experimental outcome on dilution factor. The LV model predicts that increasing dilution will favor faster-growing species over slower-growing ones. If fast growers dominate at low dilution factors, though, no changes in outcome will be expected. Changes in outcome are therefore most dramatic when slow growers are strong competitors at low dilution, exhibiting a tradeoff between growth rate and competitive ability. **a** This tradeoff was pervasive in our system: slower growth rates resulted in higher competitive scores at the lowest dilution factor. Growth rate was calculated with OD600 measurements of the time taken for monocultures to reach a threshold density within the exponential phase; error bars represent the SEM of replicates (*n* = 21, per species) (Supplementary Fig. 3). Competitive score was calculated by averaging fraction of a given species across all pairwise competitive outcomes; error bars were calculated by bootstrapping, where replicates of mean experimental outcomes of a given pair were sampled 5000 times with replacement (*n* = 34, per species, per dilution factor). **b** The competitive scores in **a** are extended to all dilution factors. The slowest grower's score monotonically decreases with dilution, while the fast growers' scores increase, and an intermediate grower peaks at intermediate dilution factor. A similar pattern was seen in data from experiments in a complex growth medium (Supplementary Fig. 7). **c** At high dilution factors, the order of scores is reversed. **d** At low dilution factors 10 and $10^2$, competitive ability is negatively correlated with growth rate; the correlation becomes positive above dilution factor $10^3$. Error bars are the standard error coefficients given by the linear regression function lm in R. Source data are provided as a Source Data file

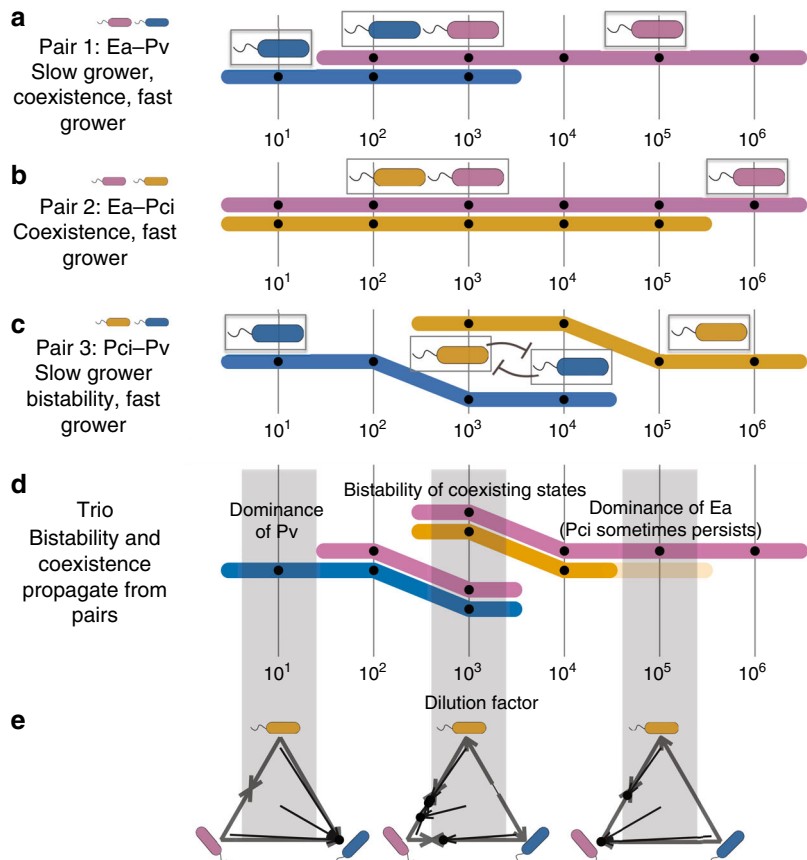

**Fig. 5** Coexistence and bistability propagate from pair to trio, as predicted by assembly rules. **a–c** Subway maps show pairwise outcome trajectories across changing dilution factor (DF), as explained in Figs. 1 and 3. The fast grower's line is always plotted above the slow grower's line. Of the three pairs that make up the community *Ea–Pci–Pv*, two are coexisting (**a**, **b**) and one is bistable (**c**). **d** The pairwise assembly rules state that a species will survive in a community if it survives in all corresponding pairs. At DF 10, *Ea* and *Pci* coexist, but both are excluded by *Pv*. The rules correctly predict that *Pv* will dominate in the trio. Because both species can be excluded in a bistable pair, a bistable pairwise outcome propagates to the trio as more than one allowed state. Each of the bistable species can be seen separately coexisting with *Ea* at DF $10^3$, as they do in pairs. The assembly rules failed at DF $10^5$ for three out of four starting conditions: *Pci* usually goes extinct when it should coexist with *Ea*. **e** Three-species competition results are shown in simplex plots. Arrows begin and end at initial and final fractions, respectively. Edges represent pairwise results, and black dots represent trio results

**Table 1 Errors of pairwise assembly rules are much lower than monoculture prediction errors**

| Prediction type | Mean error/max. error, trios | Mean error/max. error, quads | Overall |
|---|---|---|---|
| Pairwise outcomes | 0.09 (0.02) | 0.19 (0.02) | 0.14 (0.01) |
| Carrying capacities (monocultures) | 0.40 (0.03) | 0.43 (0.03) | 0.41 (0.02) |
| Random predictions | 0.46 (0.03) | 0.47 (0.03) | 0.47 (0.02) |

We made quantitative predictions of the relative fractions in multispecies competition outcomes using both monoculture carrying capacities as well as the pairwise assembly rules. Mean errors of both types of predictions are shown, as well as mean error for random predictions, for comparison. Errors of quantitative predictions are the L2 norm of the distance between predicted fixed point and observed fixed point (see "Methods" and Supplementary Fig. 2). The values shown are mean error normalized by the maximum error. Errors, shown in parentheses, are SEM of replicates ($n = 118$ for trios, $n = 96$ for quads; outcomes of different starting fractions of the same biological replicate were averaged before measuring error, and in the case of bistability, the smaller of the two errors was chosen after averaging outcomes from different starting fractions ending at the same final state)

To determine whether our analysis of community shifts under mortality is more broadly applicable, we combined our five species into various three- and four-species subsets, similar to the *Ea–Pci–Pv* competition (Fig. 5). In total, we competed five three-species communities and three four-species communities at all six dilution factors (see Supplementary Fig. 9 for examples, as well as for result of five-species coculture). Overall, a quantitative generalization of our assembly rules (see "Methods") predicted the equilibrium fractions with an error of 14%, significantly better than the 41% error that results from predictions obtained from monoculture carrying capacity (Table 1, Supplementary Fig. 2). Assembly rule prediction error does increase with increasing

community size, however, particularly in the case of the five-species community (Supplementary Table 1, Supplementary Fig. 9), which may be due to slow equilibration or infrequent coexistence of more than two species. These results indicate that pairwise outcomes are good predictors of simple multispecies states in the presence of increased mortality.

## Discussion

The question of how community composition will change in a deteriorating environment is essential, as climate change, ocean acidification, and deforestation infringe upon many organisms'

habitats, increasing mortality either directly, by decimating populations, or indirectly, by making the environment less hospitable to them. We used an experimentally tractable microbial microcosm to tune mortality through dilution rate and found a pervasive tradeoff between growth rate and competitive ability (Fig. 4). This tradeoff causes slow growers to outcompete fast growers in high-density, low-dilution environments. Increasing mortality favors fast growers, in line with model predictions. We observed coexistence and bistability at intermediate dilution factors in pairwise experiments (Fig. 3) and found that such coexistence and bistability propagated up to three- and four-species communities (Fig. 5). Coexistence was more common than bistability, which is in line with expectations of optimal foraging theory[1]. We were able to explain seemingly complicated three-species states (Fig. 1) with pairwise results, which traversed all possible outcomes allowed by the two-species model.

The success of the simple pairwise assembly rules[28] in predicting the states of three- and four-species communities (Table 1, Supplementary Fig. 2) is in line with recent microbial experiments suggesting that pairwise interactions play a key role in determining multispecies community assembly[28,31] and community-level metabolic rates[32]. In contrast, some theory and empirical evidence supports the notion of pervasive and strong higher-order interactions[33–36]. Our results provide support for a bottom–up approach to simple multispecies communities and show that pairwise interactions alone can generate multispecies states that appear nontrivial. Prediction errors do increase with increasing community size, however, as can be seen in the case of the five-species community (Supplementary Table 1, Supplementary Fig. 9).

The aforementioned tradeoff made for striking transitions in the communities that we studied. Without the tradeoff, the model would be less useful—if a fast grower outcompetes a slow grower at low dilution rates, the model predicts no change in outcome at higher dilution rates. Our results at low dilution are consistent with previous experimental evidence of a tradeoff between growth and competitive ability among different mutants of the same bacterial strain[37] and between different species of protists[38,39]. Other examples of this tradeoff include antibiotic resistance, which imposes a fitness cost on bacteria despite its clear competitive benefit[40], and seed size in plants; plants that produce larger seeds necessarily produce fewer of them but were found to be more competitive in seedling establishment[41,42]. The high competitive score of slow growers in our system in low-dilution environments, together with the result that increasing dilution favors fast growers, provides a case study for how a unimodal diversity–disturbance relationship can occur in a microbial community, a phenomenon that has previously been observed[43–45].

The exact mechanism for the competitive ability of the slow growers in our system cannot be fully explained. Monoculture pH levels were similar for all species (~6.2–6.5), ruling out the possibility that slow growers move the pH to a level not hospitable to the fast growers. Supernatant experiments, in which we grew each species in the filtered spent media of other species, showed inhibition of some fast growers (Pp, Pci) by some slow growers (Pa, Pv) (Supplementary Fig. 6b, d), which explains potentially three of the seven cases of slow grower dominance at low dilution factor. We also hypothesized that the tradeoff might be caused by the slow growers having relatively faster growth rates at low resource concentration (as explained below), but this hypothesis was not confirmed when tested (Supplementary Fig. 6f). In addition, in monocultures the slow growers exhibited higher lag times than the fast growers (Supplementary Fig. 5f), which would seem to be disadvantageous in low-dilution, high-density conditions where resources could be quickly consumed by a competitor with a shorter lag[46]. The reason for the tradeoff, as well as its

frequency in other systems, is a matter worthy of further investigation, in particular because natural microbial systems, such as soil communities or the gut microbiome, are characterized as having a low dilution rate[47,48].

Here we found that the LV model with added mortality provided useful guidance for how experimental competition would shift under increased dilution, but resource-explicit models may in some cases provide additional mechanistic insight[49,50]. In particular, various resource-explicit models can recapitulate the qualitative changes predicted by the LV model with added mortality. For example, the R* rule states the species that can survive on the lowest equilibrium resource concentration will dominate other species[1]. The equilibrium concentration increases with the dilution rate, thus favoring the species with the highest maximal growth rate (Supplementary Note 4, Supplementary Figs. 12 and 13). However, a species with a low maximal rate may dominate under low dilution if it can grow more efficiently at low resource concentrations. As mentioned, this hypothesis could not explain the tradeoff in our system (Supplementary Fig. 6f). Moreover, while we consider dilution to be essentially an added death rate because cells are discarded, the LV model does not include effects of the dilution process that could differentiate it from mere mortality. Previous experimental work has shown that dilution can modulate concentrations of oxygen[51,52] and phosphate[45] in the environment, leading to changes in microbial community composition. Further work is necessary to explore the circumstances in which phenomenological or resource-explicit models should be used[53–55] in describing serial dilution experiments.

It is also important to note that not all deteriorating environments will cause such simple and uniform increases in mortality. Antibiotics, and in particular β-lactam antibiotics, might selectively attack fast growers over slow growers[56]. Overfishing might target certain species of fish. In such cases of species-specific mortality rates, the pairwise LV model still predicts that outcomes will move along the same 45° line through the phase space but in a direction dependent on the differing rates (Supplementary Note 1). Climate change might affect growth rate rather than death rate by increasing temperature, which usually increases growth rates[57]; in this case, it is not certain whether environmental deterioration in the form of warming would favor slow growers or fast growers. An important direction for future research is to determine whether changes to the environment other than mortality/dilution will have predictable consequences for the composition of microbial communities. In this study, we have seen how a simple prediction about a simple perturbation in pairwise competition—increased mortality will favor the faster-growing species—allowed us to interpret seemingly nontrivial outcomes in simple multispecies communities.

## Methods

**Species and media**. The soil bacterial species used in this study were *Enterobacter aerogenes* (Ea, ATCC#13048), *Pseudomonas aurantiaca* (Pa, ATCC#33663), *Pseudomonas citronellolis* (Pci, ATCC#13674), *Pseudomonas putida* (Pp, ATCC#12633), and *Pseudomonas veronii* (Pv, ATCC#700474). All species were obtained from ATCC. Two types of growth media were used: one was complex and undefined, while the other was minimal and defined. All results presented in the main text are from the defined media. All species grew in monoculture in both media. The complex medium was 0.1× LB broth (diluted in water). The minimal medium was S medium, supplemented with glucose and ammonium chloride. It contains 100 mM sodium chloride, 5.7 mM dipotassium phosphate, 44.1 mM monopotassium phosphate, 5 mg/l cholesterol, 10 mM potassium citrate pH 6 (1 mM citric acid monohydrate, 10 mM tri-potassium citrate monohydrate), 3 mM calcium chloride, 3 mM magnesium sulfate, and trace metals' solution (0.05 mM disodium EDTA, 0.02 mM iron sulfate heptahydrate, 0.01 mM manganese chloride tetrahydrate, 0.01 mM zinc sulfate heptahydrate, 0.01 mM copper sulfate pentahydrate), 0.93 mM ammonium chloride, and 10 mM glucose. 1× LB broth was used for initial inoculation of colonies. For competitions involving more than two species, plating was done on 10 cm circular Petri dishes containing 25 ml of nutrient agar (nutrient broth (0.3% yeast extract, 0.5% peptone) with 1.5% agar

added). For pairwise competitions, plating was done on rectangular Petri dishes containing 45 ml of nutrient agar, onto which diluted 96-well plates were pipetted at 10 μl per well.

**Growth rate measurements**. Growth curves were captured by measuring the optical density of monocultures (OD 600 nm) in 15-min intervals over a period of ~50 h (Fig. S3). Before these measurements, species were grown in 1× LB broth overnight and then transferred to the experimental medium for 24 h. The OD of all species was then equalized. The resulting cultures were diluted into fresh medium at factors of $10^{-8}$ to $10^{-3}$ of the equalized OD. Growth rates were measured by assuming exponential growth to a threshold of OD 0.1 and averaging across many starting densities and replicates ($n = 19$ for $Pci$, $n = 22$ for all other species). This time-to-threshold measurement implicitly incorporates lag times, because a species with a time lag will take longer to reach the threshold OD than another species with the same exponential rate but no lag time. We also estimated lag times and exponential rates explicitly (Fig. S4). We used these measurements to develop an alternative to the time-to-threshold rates, which also incorporated lag time. To estimate this effective growth rate, we multiplied the exponential rate by a factor depending on lag time and time between daily dilutions (Supplementary Fig. 5b and Supplementary Note 3). This method does change growth rate estimates slightly but does not change the order of growth rates among the five species and thus the qualitative predictions of the model (Supplementary Fig. 5a, b). For this reason, we preferred to use the time-to-threshold method, because it involved only one measurement, rather than two, and had a lower error.

**Competition experiments**. Frozen stocks of individual species were streaked out on nutrient agar Petri dishes, grown at room temperature for 48 h, and then stored at 4 °C for up to 2 weeks. Before competition experiments, single colonies were picked and each species was grown separately in 50 ml Falcon tubes, first in 5 ml LB broth for 24 h and next in 5 ml of the experimental media for 24 h. During the competition experiments, cultures were grown in 500 μl 96-well plates (BD Biosciences), with each well containing a 200-μl culture. Plates were incubated at 25 °C and shaken at 400 rpm and were covered with an AeraSeal film (Sigma-Aldrich). For each growth–dilution cycle, the cultures were incubated for 24 h and then serially diluted into fresh growth media. Initial cultures were prepared by equalizing OD to the lowest density measured among competing species, mixing by volume to the desired species composition, and then diluting mixtures by the factor to which they would be diluted daily (except for dilution factor $10^{-6}$, which began at $10^{-5}$ on Day 0, to avoid causing stochastic extinction of any species). Relative abundances were measured by plating on nutrient agar plates. Each culture was diluted in phosphate-buffered saline prior to plating. For competitions involving more than two species, plating was done on 10 cm circular Petri dishes. For pairwise competitions, plating was done on 96-well-plate-sized rectangular Petri dishes containing 45 ml of nutrient agar, onto which diluted 96-well plates were pipetted at 10 μl per well. Multiple replicates of the latter dishes were used to ensure that enough colonies could be counted. Colonies were counted after 48 h incubation at room temperature. The mean number of colonies counted, per plating, per experimental condition, was 42. During competition experiments, we also plated monocultures to determine whether each species could survive each dilution factor in the absences of other species. $Pv$ went extinct in the highest two dilution factors, while $Pa$ went extinct in the highest dilution factor; all other species survived all dilution factors (Fig. 4).

**Assembly rule predictions and accuracy**. In order to make predictions about three- and four-species states, we used the qualitative and quantitative outcomes of pairwise competition. The two types of pairwise outcomes allowed for two types of predictions. First, the qualitative outcomes (dominance/exclusion, coexistence, or bistability) of the pairs were used to predict whether a species would be present or absent from a community. These outcomes are shown in "subway maps" (Supplementary Fig. 1), where the presence of a species is noted by the presence of its assigned color. Coexistence is shown by two stacked colors, and bistability is shown by two separated colors. The qualitative error rate is the percentage of species, out of the total number of species (three for trios, four for quads), that are incorrectly predicted to be present or absent (Table 1, Supplementary Fig. 2a, b). The qualitative success rate is the percentage of species that are correctly predicted as present or absent (Supplementary Fig. 2d).

Second, the quantitative outcomes of the pairs were used to predict the quantitative outcomes of three- and four-species communities. These outcomes are shown in relative fraction plots (Supplementary Fig. 1), where equilibrium points are indicated by the black dots. When two or more species coexist in pairs, the assembly rules predicts that they will coexist in multispecies communities, provided that an additional species does not exclude them. The predicted equilibrium coexisting fraction of two species is the same in a community as it is in a pair, while the fractions of more than two coexisting species are predicted with the weighted geometric mean of pairwise coexisting fractions. For example, in a three-species coexisting community, the fraction of species 1 depends on its coexisting fractions with the other two species in pairs:

$$f_1 = \left(f_{12}^{w_2} f_{13}^{w_3}\right)^{\frac{1}{w_2+w_3}}$$

where $f_{12}$ is the fraction of species 1 after reaching equilibrium in competition with species 2, $w_2 = \sqrt{f_{21} f_{23}}$ and $w_3 = \sqrt{f_{31} f_{32}}$. Finally, these predictions are normalized by setting $f_1^* = \frac{f_1}{f_1+f_2+f_3}$. The quantitative error of a particular community outcome is the distance of the predicted fractions from the observed community fractions, measured with the L2 norm. The maximum error, for any number of species, is $\sqrt{2}$, which occurs when a species that was predicted to go extinct in fact dominates:

$$\sqrt{\sum((1,0,\dots,0)-(0,1,\dots,0))^2} = \sqrt{2}.$$

To calculate the overall quantitative errors (Table 1, Supplementary Fig. 2c, Supplementary Table 1), we divided each error by $\sqrt{2}$ and took the mean.

Finally, we also predicted multispecies states using carrying capacities as measured in monocultures through colony counting (Supplementary Fig. 5c, d). We assumed that, in competition, each species would grow to a density proportionate to its carrying capacity. In other words, the monoculture prediction assumes that all species always coexist. The error from the prediction to the observed data was calculated with the L2 norm, as above.

**Statistical analysis**. The $p$ values given in Supplementary Figs. 3 and 5 were obtained using two-tailed $t$ tests. The error bars shown in the time-series plots in Figs. 1 and 3 and Supplementary Fig. 8 are the SD of the beta distribution with Bayes' prior probability:

$$\sigma = \sqrt{\frac{(\alpha+1)(\beta+1)}{(\alpha+\beta+2)^2(\alpha+\beta+3)}}.$$

Here $\alpha$ and $\beta$ are the number of colonies of two different species. In case of more than two species, $\alpha$ and $\beta$ are the number of colonies of a given species and the number of all other species' colonies, respectively.

**Reporting summary**. Further information on research design is available in the Nature Research Reporting Summary linked to this article.

## Data availability
The source data underlying Figs. 1b, 3b, f, and 4a–c, and Supplementary Figs. 1, 2, 5, and 7 are provided as a Source Data file. Access to the data is also publicly available at https://figshare.com/projects/Added_mortality_causes_universal_changes_in_microbial_community_composition/58304. A reporting summary for this article is available as a Supplementary Information file.

## Code availability
The code used for analyzing data is available from the first author upon request.

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

## Acknowledgements

We thank the members of the Gore Laboratory for critical discussions and comments on the manuscript.

## Author contributions

All the authors designed the study, discussed and interpreted the results, and wrote the manuscript. C.I.A. and V.L.A.W. carried out the experiments and performed the analysis.

## Additional information

**Competing interests:** The authors declare no competing interests.

