## [Peer Review File · Nature Communications]

Reviewers' Comments:

Reviewer #1:

Remarks to the Author:

General Assessment:

The authors present a nice set of experiments targeted at assessing how non-specific mortality influences competitive outcomes in simple microbial communities. The experiments show a diverse set of ecological states, with coexistence, bi-stability, and single-species dominating along a mortality gradient. The experiments are accompanied by a LV modeling framework that provides some mechanistic insight into how mortality alters competition between organisms with different growth rates and competitive abilities. Overall, the authors show strong evidence for a trade-off between competitive ability and growth rate in their model system. This is an elegant study with a clear message. The text is well written. The figures are great and easily digestible. The results are novel and compelling and I think this will make a nice addition to the literature. My only concern is that the authors may want to add more discussion of the literature regarding the potential interaction between dilution (i.e. 'non-specific' mortality) and environmental parameters. In prior work, dilution has been shown to change specific environmental parameters. You discuss this in a general sense when talking about growth rates vs. resource concentrations, but there are a few microcosm studies that have looked into this question that could be discussed.

Minor Comments:

1) Work from several years ago found that the degree of coexistence of *Pseudomonas fluorescens* colony morphotypes varied with dilution rate (<https://www.ncbi.nlm.nih.gov/pubmed/11140680>). Dilution disrupted the spatial structure of the system, which favored certain morphotypes over others. Later, it was shown that oxygen was the environmental variable that was being modulated by dilution (<https://www.nature.com/articles/nature01906>). Another recent bacterial microcosm study found that a feedback between dilution rate and biomass levels altered phosphate concentrations, which may have been responsible for shifts in competition along a phosphate gradient (<https://mbio.asm.org/content/7/6/e01372-16.short>). Thus, dilution may not simply be a non-specific mortality rate, but a force that modulates niche space as well. You mention that resource concentration is likely reduced at low-dilution rates, giving the superior competitor an advantage. In your experiments, did you measure potentially limiting nutrients or resources? This can be done, even in complex medium (although admittedly it's difficult). Do you have an intuition for what variables are responsible for this limitation?

2) Prior work mentioned above has also shown greater dispersion in microbial community structure at intermediate dilution rates (<https://mbio.asm.org/content/7/6/e01372-16.short>). This phenomenon appears to be relatively common in microbial communities, and your work provides a nice case-study for how this can arise.

3) Line 169: Is LV entirely phenomenological? The model includes growth-rate parameters, which seem to have a concrete meaning/interpretation.

4) What volumes did you work with when culturing these communities? Were they mixed during growth?

Reviewer #2:

Remarks to the Author:

This Review is provided by Wenying Shou.

I have heard Jeff talking about this work and found the results interesting. However, unlike his other papers which were easy (and a pleasure) to read, this manuscript was difficult to follow, from abstract to the main text.

I think that the confusing point is probably caused by not clearly defining "faster grower" and "competitiveness" at the beginning of paper. The statement "faster growers are less competitive" is confusing without proper setup.

Perhaps, one could write something like this:

We define "faster grower" as the growth rate of cells at low density in monocultures. In our coculture experiments, cells were grown from low density to high density in the span of 24 hrs. How competitive one strain is against another will rely on multiple phenotypes in monoculture and coculture at different starting ratios: the duration of lag phase, growth rate in exponential phase, the maximal density a strain can reach, and viability at high cell density. Thus, we define competitiveness as xxx. Some of the supplementary figures could be referenced here.

I would also just say "coculture experiments" (instead of "competition experiments"), since other interactions can go on in your communities. If one species releases a toxin that inhibits another species' growth, then species interaction is not merely competition, but rather antagonistic.

It is not clear why the slower grower won when there was no dilution. Authors touched on that point in Lines 194-195 and in discussions and supplementary figures, but the answer is not as satisfying as one would like to have. Was carrying capacity measured as colony forming units or total cell counts? Have authors tracked the dynamics of two species in monocultures and coculture at a much higher temporal resolution than once per day? But at the least, this part needs to be advanced to the beginning to reduce confusion.

Examples of confusing narratives:

Line 53: needs to explain better

Line 54-59: need better justification of novelty of this study. For example, you can say: "although previous experiments have shown xxx, there was no theory behind experiments" etc.

Line 95: how many replicates per condition?

Line 276: I cannot judge the accuracy of this statement: Fig S6 needs to be replotted so that different conditions are clearly distinguishable. Use different line styles may help.

Figure 1 legend: should be dilution factor (not dilution rate) to be consistent with the rest of paper. Dilution rate can also mean the nutrient supply rate, which is confusing in this context. Presumably, you measured species abundance once per day, so data points should be discrete and should be indicated by symbols.

Figure 2 legend: "If it is a strong competitor, the slow grower can exclude the fast grower." What is "it"?

This paper will likely benefit from a hypothetical numerical example, like what Stan Leibler did in his Simpson's paradox paper. Note that Leibler's case is actually a lot more straightforward than your case.

Minor:

Fig 4: you could mark "low dilution" and "high dilution" in the left and right panels, respectively.

Reviewer #3:

Remarks to the Author:

In this manuscript, the authors report on how microbial communities composed of up to five species respond to growth-dilution experiments, and how predictions on the competitive outcome depend on the dilution rate in serial batch cultures. They combine experiments with modeling and show that depending on dilution rate, they can predict whether slow-growing or fast-growing species will dominate, or whether co-existence between species or bistable outcomes will be observed. The paper also builds on previous work by showing how experimentally measuring pairwise interactions can help predict outcomes with more species.

In the summary above I specifically mention growth-dilution experiments, since I am not very fond of the emphasis on mortality in the paper. I see why such a parallel may be made, as part of the population is disappearing at regular intervals. In reality, of course, death does not occur at discrete time intervals. The authors do however show in SI that discrete or continuous dilutions should not alter the qualitative behavior of their system. I still find the formulation misleading. In the abstract, "environmental harshness" makes me as a reader think of toxic environments or extreme temperatures, which is not at all what follows. Furthermore, mortality can affect a community in other ways than just a drop in population size. For example, cell death may lead to the release of DNA that can be taken up by cells, or compounds that can serve as nutrients or toxins. Finally, it is not clear that mortality would sample a population through a uniform distribution, as occurs in dilutions. In short, I would prefer if the link to mortality was made as a point in the discussion rather than the title of the paper and the main focus in the abstract and the conclusion. Instead, more emphasis could be put on how we set up bacterial culture experiments (batch versus chemostat and the role of dilution rate). At the very least, the abstract should mention that these are dilution experiments.

The other aspect that I find a bit unsatisfying is the explanation of why slow growing species dominate at low dilutions. It is intuitive that at high dilution rates fast growers dominate – the slower growers would simply not have enough time to reach a large enough population size to compensate for the dilution rate. But the domination of slower growers is less expected and in my opinion more interesting. Eventually, this is brought up in the discussion, leaving the issue as an open question. This is despite there being evidence that slower growers produce more toxic supernatants (Figure S6): both Pp and Pci seems to grow quite a lot worse in supernatants of Pa or Pv. I would say the same by eye for Pv in Pa. Given these data, I find the following statement unconvincing: "fast growers were placed in slow growers' filtered spent media showed little or no inhibition of growth compared to controls". The caption of Figure S6 also says that "antagonism may explain some but not all of dominance of slow growers in low-dilution, high-density conditions". Wouldn't it be possible to test this statistically? In SI, there is also discussion of Tilman and resource explicit models. Are the authors suggesting that this may instead explain why slower growers dominate at low dilution? I think this connection should be made clearer.

Another concern of mine is how sometimes data for one or two species is missing. For example, in the supernatant experiment of Figure S6, why is Ea missing? Figure S7 is also missing data for Pci. I also think it would be useful to include the data of the three- and four-species competitions that is summarized in Figure S2 (include all triangles or all subway growth curves somewhere). Also, it is curious that the authors did not try a five species competition.

Finally, I think the paper is really missing the mono-culture dilutions. Surely many of these species would go extinct anyway at high dilution rates. I expect that may also be a good predictor of community dynamics. As I understand it, when using monocultures to predict outcomes, the data does not include dilutions, but single growth curves and their carrying capacities. As pointed out, if extinctions are impossible in this monoculture model, then it is not surprising that it does not predict

community dynamics well. The monoculture prediction model should instead include data from single-species dilution experiments. These data would also give an idea of the role of the presence of the different species in accelerating or decelerating extinctions in the community.

Despite these criticisms, I find the paper to be an important contribution to the field, by exploring how an important environmental factor can shape communities and alter survival of different species, whether through competition or not. I find the use of the Lotka-Volterra model to capture these dynamics to be very useful and it is impressive how the phenomenological model can capture the dynamics so well (Fig. 4 is fantastic). It would be nice to see, however, whether a resource-explicit model would fill the remaining gaps. Overall, I think this is a very nice paper, but it still leaves some questions for me and could be more complete.

Minor comments:

- Figure 1 is difficult to understand. I find the subway map unintuitive. Also, where are the remaining 2 sets of data starting with different fractions?
- In Figure 2C: what determines where the line intersects with the axes? I don't follow the intuition of what the different quadrants tell you about the outcome of competition. The SI did not really help.
- Fig. S5: Not sure about those negative correlations. Can you include statistics?
- Why do you think that in defined medium you get better predictions?
- Line 155: "at which it could have survived in the absence of a competitor". Where is that data? Why do we not see monoculture dilution rates to compare?
- Can the model predict the ratio at which two species will co-exist? How well does it do?
- If you have an explicit model, why not fit it to the case that doesn't work with LV (lines 174ff) and see whether it does better?
- Why is it so hard to get coexistence of all three or four? Do you think it has something to do with medium complexity? Do you see a different outcome in the complex medium? That might be interesting to highlight.

We thank the editor and referees for their careful reading of the manuscript. We believe that the changes suggested by the reviewers have greatly strengthened the revised manuscript.

The primary changes to the manuscript include:

1. We have tried to clarify our abstract and introduction to make the narrative easier to follow. This includes an early and clear explanation of what we mean by a “fast grower” and a “strong competitor.” Additionally, we now refer to our experiments as “cocultures” rather than “competitions.”
2. We explain the tradeoff between growth rate and competitive ability in the introduction, and address it further in subsequent sections. We give a better explanation, including new experimental data, exploring how a slow grower might outcompete a fast grower at low dilution.
3. In response to reviewers’ concerns about the differences between mortality and dilution, we clarify, in the abstract, that “mortality” refers to dilution in our experiments. We expand our discussion of the differences between dilution and mortality in the discussion section, and explain that although we view dilution as effectively an added mortality rate, there are potential differences. We address the possibility of species-specific mortality rates, rather than a constant global rate, in the discussion and supplementary sections. Overall, we prefer not to completely replace “mortality” with “dilution,” because we believe that many of the conclusions reached in our manuscript may also be of relevance to researchers outside of microbiology.

Note: Line number references below are to those of the final version of the manuscript. The line numbers are inaccurate in the manuscript with highlighted changes.

Reviewer #1 (Remarks to the Author):

General Assessment:

The authors present a nice set of experiments targeted at assessing how non-specific mortality influences competitive outcomes in simple microbial communities. The experiments show a diverse set of ecological states, with coexistence, bi-stability, and single-species dominating along a mortality gradient. The experiments are accompanied by a LV modeling framework that provides some mechanistic insight into how mortality alters competition between organisms with different growth rates and competitive abilities. Overall, the authors show strong evidence for a trade-off between competitive ability and growth rate in their model system. This is an elegant study with a clear message. The text is well written. The figures are great and easily digestible. The results are novel and compelling and I think this will make a nice addition to the literature.

We thank reviewer #1 for their positive summary of our manuscript.

My only concern is that the authors may want to add more discussion of the literature regarding the potential interaction between dilution (i.e. 'non-specific' mortality) and environmental parameters. In prior work, dilution has been shown to change specific environmental parameters. You discuss this in a general sense when talking about growth rates vs. resource concentrations, but there are a few microcosm studies that have looked into this question that could be discussed.

Minor Comments:

1) Work from several years ago found that the degree of coexistence of *Pseudomonas fluorescens* colony morphotypes varied with dilution rate (<https://www.ncbi.nlm.nih.gov/pubmed/11140680>). Dilution disrupted the spatial structure of the system, which favored certain morphotypes over others. Later, it was shown that oxygen was the environmental variable that was being modulated by dilution (<https://www.nature.com/articles/nature01906>). Another recent bacterial microcosm study found that a feedback between dilution rate and biomass levels altered phosphate concentrations, which may have been responsible for shifts in competition along a phosphate gradient (<https://mbio.asm.org/content/7/6/e01372-16.short>). Thus, dilution may not simply be a non-specific mortality rate, but a force that modulates niche space as well. You mention that resource concentration is likely reduced at low-dilution rates, giving the superior competitor an advantage. In your experiments, did you measure potentially limiting nutrients or resources? This can be done, even in complex medium (although admittedly it's difficult). Do you have an intuition for what variables are responsible for this limitation?

We thank the reviewer for bringing these results to our attention. We have added some discussion of this work and cited it on lines 341-349:

“Moreover, while we consider dilution to be essentially an added death rate because cells are discarded, the LV model does not include effects of the dilution process that could differentiate it from mere mortality. Previous experimental work has shown that dilution can modulate concentrations of oxygen^{50,51} and phosphate⁴⁴ in the environment, leading to changes in microbial community composition. Further work is necessary to explore the circumstances in which phenomenological or resource-explicit models should be used^{52–54} in describing serial dilution experiments.”

Regarding the role of limiting nutrients, we have made new measurements of monoculture growth rates at low resource concentration to test the hypothesis that slow growers win at low dilution factors because the ordering of growth rates flips in saturated environments with low resource concentration, leading the slow growers to become relatively faster. This possible origin of slow grower dominance at low dilution factor does not appear to apply to our species, as can be seen in the panel we have added to Supplementary Figure 6. In particular, the ordering of growth rates does not flip at low resource concentration, except for the rank of *Pp*. At most, this data could explain the exclusion of *Pp* by *Ea* or *Pci* at low dilution factor, but in fact these species coexist at low dilution factor. We have not measured the concentration of resources during growth.

2) Prior work mentioned above has also shown greater dispersion in microbial community structure at intermediate dilution rates (<https://mbio.asm.org/content/7/6/e01372-16.short>). This phenomenon appears to be relatively common in microbial communities, and your work provides a nice case-study for how this can arise.

We appreciate the reviewer’s suggestions—we have mentioned and cited this and two other papers on lines 307-311: “The high competitive score of slow growers in our system in low-dilution environments, together with the result that increasing dilution favors fast growers, provides a case study for how a unimodal diversity-disturbance relationship can occur in a microbial community, a phenomenon that has previously been observed.^{42–44}”

3) Line 169: Is LV entirely phenomenological? The model includes growth-rate parameters, which seem to have a concrete meaning/interpretation.

This is true, but these growth rates are maximal, and the model’s assumption of logistic growth is phenomenological because it implies but does not specifically model a limited amount of resources. Additionally, the interactions are summarized with competition coefficients, which could describe anything from resource-mediated competition to direct cell-to-cell warfare.

4) What volumes did you work with when culturing these communities? Were they mixed during growth?

We used 200ul volumes at 25 degrees, shaking at 400 rpm (see Methods section, lines 420-421).

Reviewer #2 (Remarks to the Author):

This Review is provided by Wenying Shou.

I have heard Jeff talking about this work and found the results interesting. However, unlike his other papers which were easy (and a pleasure) to read, this manuscript was difficult to follow, from abstract to the main text.

We apologize that this work was not as clear as we had intended. We hope that clarifying the points below will help readers.

I think that the confusing point is probably caused by not clearly defining "faster grower" and "competitiveness" at the beginning of paper. The statement "faster growers are less competitive" is confusing without proper setup.

Perhaps, one could write something like this:

We define "faster grower" as the growth rate of cells at low density in monocultures. In our coculture experiments, cells were grown from low density to high density in the span of 24 hrs. How competitive one strain is against another will rely on multiple phenotypes in monoculture and coculture at different starting ratios: the duration of lag phase, growth rate in exponential phase, the maximal density a strain can reach, and viability at high cell density. Thus, we define competitiveness as xxx. Some of the supplementary figures could be referenced here.

We appreciate this suggestion. We have added something like it in lines 65-77 of the introduction section of the new text:

"... 1) increased mortality favors the fast grower, and can reverse the "winner," or the only remaining species at the end of the experiment, from slow grower to fast grower, and 2) at intermediate dilution rates, either coexistence or bistability occurs, where from many starting fractions, the two species' abundances either converge to a stable fraction or diverge to either species winning, respectively. **We measure species' growth rates by growing cells from low density in monoculture; fast growers reach a threshold density more quickly than slow growers. We define the competitive ability of a species as its average fraction after being cocultured in pairs with each of the other species for multiple dilution cycles.** Interestingly, we find that a pervasive tradeoff between growth rate and competitive ability in our system favors slow growers in high-density, low-dilution environments, leading to striking changes in outcomes as mortality increases."

We hope that this clarifies our definitions and sets up the narrative for the rest of the paper. We prefer not to refer to the supplementary figure showing details like time lags and carrying capacities until the discussion section.

I would also just say "coculture experiments" (instead of "competition experiments"), since other interactions can go on in your communities. If one species releases a toxin that inhibits another species' growth, then species interaction is not merely competition, but rather antagonistic.

We agree that referring to these as coculture experiments is more general, so we have made this change throughout the manuscript.

It is not clear why the slower grower won when there was no dilution. Authors touched on that point in Lines 194-195 and in discussions and supplementary figures, but the answer is not as satisfying as one would like to have. Was carrying capacity measured as colony forming units or total cell counts? Have authors tracked the dynamics of two species in monocultures and coculture at a much higher temporal resolution than once per day? But at the least, this part needs to be advanced to the beginning to reduce confusion.

Carrying capacity was measured as colony-forming units. We did not track temporal dynamics of the cocultures throughout the day, but to better address the question of why slow growers win at low dilution, we have performed

new experiments—first, we reproduced the growth inhibition of *Pp* and *Pci* (fast growers) by *Pa* and *Pv* (slow growers), respectively (as seen in Supplementary Figure 6):

There is some evidence of *Pp* inhibition by *Pv*, but only in one of the replicates. This test therefore explains potentially two or three of the seven instances of slow grower dominance at low dilution factor. *Ea*, another fast grower, was not inhibited by slow growers' supernatant. We also measured pH of the supernatants, and found similar values for all (~6.2 – 6.5), ruling out the possibility that slow growers move the pH to a level not hospitable to the fast growers.

Second, we made new measurements of monoculture growth rates at low resource concentration to test the hypothesis that slow growers win at low dilution factors because the ordering of growth rates flips in saturated environments with low resource concentration, leading the slow growers to become relatively faster. This possible origin of slow grower dominance at low dilution factor does not appear to apply to our species, as can be seen in the panel we have added to Supplementary Figure 6. In particular, the ordering of growth rates does not flip at low resource concentration, except for the rank of *Pp*. At most, this data could explain the exclusion of *Pp* by *Ea* or *Pci* at low dilution factor, but in fact these species coexist at low dilution factor.

We conclude that toxins or other compounds in the supernatant explain some, but not all, of slow growers' competitive ability, but that the matter remains an open question. We think that this expanded discussion (paragraph 4 of discussion section), as well as the better explanations of growth rate and competitive ability (mentioned above) in the introduction will bring up the issue more clearly and earlier in the paper.

Examples of confusing narratives:
Line 53: needs to explain better

We have added "as mortality increases" to clarify that the outcome reversal occurs because increasing mortality favors faster growers (lines 49-52 of new text):

“it is predicted that such a global mortality rate will favor the faster-growing species in pairwise coculture, potentially reversing outcomes from dominance of the slow grower to dominance of the fast grower^{1,20,21} as mortality increases.”

Line 54-59: need better justification of novelty of this study. For example, you can say: “although previous experiments have shown xxx, there was no theory behind experiments” etc.

We have added the following justification to lines 57-60 of the new text:

“Missing from the literature is a systematic study that probes both of these predictions with an array of pairwise coculture experiments across a range of dilution rates.”

Line 95: how many replicates per condition?

The experiment was replicated three times, and outcomes changed slightly in other replicates—we have expanded the data on this in Supplementary Figure 8 (also included are the last two starting conditions from Fig. 1):

Line 276: I cannot judge the accuracy of this statement: Fig S6 needs to be replotted so that different conditions are clearly distinguishable. Use different line styles may help.

We also thought that the statement was too dismissive of the potential effect of slow growers' supernatant on fast growers, and have changed it (lines 315-319 of new text):

“Supernatant experiments, in which we grew each species in the filtered spent media of other species, showed inhibition of some fast growers (*Pp*, *Pci*) by some slow growers (*Pa*, *Pv*) (Supplementary Figure 6b, d), which explains potentially three of the seven cases of slow grower dominance at low dilution factor.”

We used a colorblind palette for the different lines, which was also used to assign colors to the five species throughout the paper. We used it here for continuity, but please let us know if it is not adequate (see right):

Figure 1 legend: should be dilution factor (not dilution rate) to be consistent with the rest of paper. Dilution rate can also mean the nutrient supply rate, which is confusing in this context. Presumably, you measured species abundance once per day, so data points should be discrete and should be indicated by symbols.

We have changed this to “dilution factor.” We also added data points and error bars to Figure 1:

Figure 1: Increasing dilution causes striking shifts in a three-species community. A) To probe how added mortality changes community composition, we cocultured three soil bacteria over a range of dilution factors. Cells were inoculated and allowed to grow for 24 hours before being diluted into fresh

Figure 2 legend: “If it is a strong competitor, the slow grower can exclude the fast grower.” What is “it”?

“it” = the slow grower. The text has been changed to clarify this:

“If the slow grower is a strong competitor, it can exclude the fast grower.”

This paper will likely benefit from a hypothetical numerical example, like what Stan Leibler did in his Simpson's paradox paper. Note that Leibler's case is actually a lot more straightforward than your case.

We agree that our results are not straightforward, so we have added a concrete numerical example, as suggested, in Figure 2C:

Minor:

Fig 4: you could mark "low dilution" and "high dilution" in the left and right panels, respectively.

Done.

Reviewer #3 (Remarks to the Author):

In this manuscript, the authors report on how microbial communities composed of up to five species respond to growth-dilution experiments, and how predictions on the competitive outcome depend on the dilution rate in serial batch cultures. They combine experiments with modeling and show that depending on dilution rate, they can predict whether slow-growing or fast-growing species will dominate, or whether co-existence between species or bistable outcomes will be observed. The paper also builds on previous work by showing how experimentally measuring pairwise interactions can help predict outcomes with more species.

In the summary above I specifically mention growth-dilution experiments, since I am not very fond of the emphasis on mortality in the paper. I see why such a parallel may be made, as part of the population is disappearing at regular intervals. In reality, of course, death does not occur at discrete time intervals. The authors do however show in SI that discrete or continuous dilutions should not alter the qualitative behavior of their system. I still find the formulation misleading. In the abstract, "environmental harshness" makes me as a reader think of toxic environments or extreme temperatures, which is not at all what follows. Furthermore, mortality can affect a community in other ways than just a drop in population size. For example, cell death may lead to the release of DNA that can be taken up by cells, or compounds that can serve as nutrients or toxins. Finally, it is not clear that mortality would sample a population through a uniform distribution, as occurs in dilutions. In short, I would prefer if the link to mortality was made as a point in the discussion rather than the title of the paper and the main focus in the abstract and the conclusion. Instead, more emphasis could be put on how we set up bacterial culture experiments (batch versus chemostat and the role of dilution rate). At the very least, the abstract should mention that these are dilution experiments.

We thank Reviewer 3 for the very thoughtful feedback. Our primary motivation in this project was to explore how a species-independent mortality would alter the composition of a multispecies community, as opposed to other work in our group which directly explored the role of cell death (Ratzke et. al., 2018). As a concrete manifestation of this simple form of mortality, we chose microbial microcosms with discrete dilution. However, we agree that it is better to be more explicit regarding how the experiments were done, so we have explained that we tune mortality via dilution in the abstract (lines 11-13 of new text):

"Here we combine theory and laboratory microcosms to predict how simple microbial communities will change under added mortality, controlled by varying dilution."

We also changed other mentions of "mortality" in the abstract to "dilution" (lines 17, 18). We prefer not to change the instance of "mortality" in the title to "dilution," because we believe our results may be relevant to researchers outside of microbiology, as well as to theorists who are not familiar with bacterial culture experimental protocols.

In addition, in the Discussion section we have added a further comment on how dilution can differ from mortality on lines 341-346:

"Moreover, while we consider dilution to be essentially an added death rate because cells are discarded, the LV model does not include effects of the dilution process that could differentiate it from mere mortality. Previous experimental work has shown that dilution can modulate concentrations of oxygen^{50,51} and phosphate⁴⁴ in the environment, leading to changes in microbial community composition."

To address the issue that mortality would not sample a population through a uniform distribution, we have added a sentence in the Discussion section (lines 353-356):

"In such cases of species-specific mortality rates, the pairwise LV model still predicts that outcomes will move along the same 45° line through the phase space, but in a direction dependent on the differing rates (Supplementary Note 1)"

Here is the newly added discussion in the SI to which this sentence refers:

"While a global death rate of δ leads to the simple prediction that the fast grower is favored, it is not the most realistic scenario. In reality, different species may be affected by different added death rates. In this case, the expression for the competition coefficients becomes:

$$\tilde{\alpha}_{ij} = \alpha_{ij} \frac{1 - \frac{\delta_j}{r_j}}{1 - \frac{\delta_i}{r_i}}$$

Taking the log of the above expression results in addition of a term to $\log \alpha_{ij}$, the same term which will be subtracted from $\log \alpha_{ji}$. The outcomes will therefore still move along the same 45° line through the phase

space, although not necessarily at the same rate or in the same direction. Added mortality will favor the faster grower if the following condition is met:

$$\frac{\delta_s}{\delta_f} > \frac{r_s}{r_f}$$

We therefore see that the fast grower can still be favored if it is killed at a higher rate (as in the case of β -lactam antibiotics, which target faster growers by inhibiting cell wall biosynthesis). Furthermore, the growth/competition tradeoff at low dilution is not required to observe outcome changes if the slow grower is selectively targeted; in this case, the trajectory would move from fast grower winning at low mortality, to coexistence or bistability at intermediate mortality, to the slow grower winning at high mortality.”

The other aspect that I find a bit unsatisfying is the explanation of why slow growing species dominate at low dilutions. It is intuitive that at high dilution rates fast growers dominate – the slower growers would simply not have enough time to reach a large enough population size to compensate for the dilution rate. But the domination of slower growers is less expected and in my opinion more interesting. Eventually, this is brought up in the discussion, leaving the issue as an open question. This is despite there being evidence that slower growers produce more toxic supernatants (Figure S6): both Pp and Pci seems to grow quite a lot worse in supernatants of Pa or Pv. I would say the same by eye for Pv in Pa. Given these data, I find the following statement unconvincing: “fast growers were placed in slow growers’ filtered spent media showed little or no inhibition of growth compared to controls”. The caption of Figure S6 also says that “antagonism may explain some but not all of dominance of slow growers in low-dilution, high-density conditions”. Wouldn’t it be possible to test this statistically? In SI, there is also discussion of Tilman and resource explicit models. Are the authors suggesting that this may instead explain why slower growers dominate at low dilution? I think this connection should be made clearer.

We agree that the supernatant data does suggest that a high density of (some) slow growers may inhibit the growth of (some) fast growers, and that we did not emphasize this observation enough. We have reproduced these findings in a new experiment and included the new data in Supplementary Figure 6 (left column):

We have also changed the language in the Discussion section to explain that some inhibition may be explained (lines 315-319):

“Supernatant experiments, in which we grew each species in the filtered spent media of other species, showed inhibition of some fast growers (*Pp*, *Pci*) by some slow growers (*Pa*, *Pv*) (Supplementary Figure 6b, d), which explains potentially three of the seven cases of slow grower dominance at low dilution factor.”

It is important to stress that not all fast growers' growth rates suffered in slow growers' supernatant. *Ea*, which loses to *Pa* and *Pv* at low dilution factor, grows about as well or better in their supernatants as in its own (Supplementary Figure 6; see right). We believe this is because the pseudomonad species (*Pa*, *Pci*, *Pp*, *Pv*) all prefer to consume the potassium citrate in the growth medium instead of the glucose. *Ea*, on the other hand, consumes the glucose, and when we replenish the carbon sources in the supernatants, it benefits from an increase in glucose in the pseudomonad supernatants.

This brings us to the question of resource-explicit models, and why we mention them. We included the discussion in Supplementary Note 4 to show that the Tilman model can give the same qualitative result as the LV model with added mortality: if the slow grower wins at low dilution, the outcome can change to coexistence or bistability, and then to fast grower winning, as dilution increases. We have changed the title of Supplementary Note 4 to emphasize this:

“Qualitative predictions of resource-explicit models recapitulate LV model’s qualitative predictions”

We also did mean to suggest that our slow growers might dominate at low dilution because they are relatively faster growers at low resource concentrations—this is the tradeoff required in the Tilman model for the slow grower to win at low dilution. To check this hypothesis, we performed a new monoculture experiment in our defined growth medium with low resource concentrations. We have added the data to Supplementary Figure 6, which shows that the growth rates do not change order, and that the hypothesis was not confirmed (see right). This means that it is hard to explain why fast-grower *Ea* loses to *Pv* and *Pa* at low dilution, and the issue is left as a partially open question (lines 319-322):

“We also hypothesized that the tradeoff might be caused by the slow growers having relatively faster growth rates at low resource concentration (as explained below), but this hypothesis was not confirmed when tested (Supplementary Figure 6f).”

Another concern of mine is how sometimes data for one or two species is missing. For example, in the supernatant experiment of Figure S6, why is *Ea* missing? Figure S7 is also missing data for *Pci*. I also think it would be useful to include the data of the three- and four-species competitions that is summarized in Figure S2 (include all triangles or all subway growth curves somewhere). Also, it is curious that the authors did not try a five species competition.

We agree that it was strange to not include *Ea* in the supernatant experiments. This experiment was done once when *Ea* supernatant was not available. We have repeated the experiment with all five species in all five supernatants, and updated Supplementary Figure 6 (see above for larger images of some panels):

Pci is missing from Supplementary Figure 7 because initial experiments in the complex medium included only four species.

We have added more multispecies results to the supplement (Supplementary Figure 9), including the five-species experiment, which has a mean error/max error equal to 0.31:

Finally, I think the paper is really missing the mono-culture dilutions. Surely many of these species would go extinct anyway at high dilution rates. I expect that may also be a good predictor of community dynamics. As I understand it, when using monocultures to predict outcomes, the data does not include dilutions, but single growth curves and their carrying capacities. As pointed out, if extinctions are impossible in this monoculture model, then it is not surprising that it does not predict community dynamics well. The monoculture prediction model should instead include data from single-species dilution experiments. These data would also give an idea of the role of the presence of the different species in accelerating or decelerating extinctions in the community.

During the competition experiments, we plated monocultures at all dilution factors to determine survival or extinction, but we did not count colonies. We decided not to do this, because while relative abundances are highly reproducible from experiment to experiment (e.g. coexisting fractions), we have found that absolute abundances are highly variable in colony-counting assays. We preferred to depend upon qualitative data about which species survive at

which dilution factors. We have added a note to Fig. 4 to show that *Pv* goes extinct in monoculture at the highest two dilution factors, and that *Pa* goes extinct at the highest dilution factor:

We have also added an explanation of this to the Methods section (lines 436-440):

“During competition experiments, we also plated monocultures to determine whether each species could survive each dilution factor in the absence of other species. *Pv* went extinct in the highest two dilution factors, while *Pa* went extinct in the highest dilution factor; all other species survived all dilution factors (Fig. 4).”

Using our carrying capacity estimates for each species in the absence of dilution, we have modeled the effect of dilution on carrying capacity in what we call an “effective” carrying capacity:

$$K_{effective} = K \left(1 - \frac{\delta}{r}\right)$$

We used effective carrying capacities to predict outcomes, in addition to unmodified carrying capacities. We did not include the modified carrying capacity predictions in Table 1, but we did include them in Supplementary Figure 2. The “modified” monoculture prediction errors, shown as the middle two bars in the plot to the right, are not as bad as those for carrying capacity alone (left two bars) but are significantly worse than the errors we get when incorporating the pairwise outcomes (right two bars).

Despite these criticisms, I find the paper to be an important contribution to the field, by exploring how an important environmental factor can shape communities and alter survival of different species, whether through competition or not. I find the use of the Lotka-Volterra model to capture these dynamics to be very useful and it is impressive how the phenomenological model can capture the dynamics so well (Fig. 4 is fantastic). It would be nice to see, however, whether a resource-explicit model would fill the remaining gaps. Overall, I think this is a very nice paper, but it still leaves some questions for me and could be more complete.

As mentioned above, the resource model discussion was included to show that it can make the same qualitative predictions as the Lotka-Volterra model, albeit with more parameters. As we explain below, fitting data to the Lotka-Volterra model was not very successful, and we found the qualitative predictions of the model to be more useful.

Minor comments:

- Figure 1 is difficult to understand. I find the subway map unintuitive. Also, where are the remaining 2 sets of data starting with different fractions?

We hope that the subway map is easier to understand with the outcomes labeled in Fig. 1:

We have provided the remaining 2 sets of data in Supplementary Figure 8, as well as data from when the experiment was reproduced two other times:

- In Figure 2C: what determines where the line intersects with the axes? I don't follow the intuition of what the different quadrants tell you about the outcome of competition. The SI did not really help.

We have added a sentence to the main text (lines 138-140) and the SI to clarify how the outcome depends upon the alphas:

“Stable coexistence occurs when both $\tilde{\alpha}$ coefficients are less than one, bistability when both are greater than one, and dominance/exclusion when only one coefficient is greater than one.”

As mortality is added, the outcome changes, moving along the line through the phase space. The slope of the line is always -1, so the starting point determines where the line intersects the axis. The starting point is the set of competition coefficients in the absence of growth rate or death rate. We have changed this figure to include two numerical examples, as suggested by Reviewer #2:

- Fig. S5: Not sure about those negative correlations. Can you include statistics?

Statistics have been added to show which correlations are significant (see right). Additionally, the title of the figure has been changed to “Growth rate is weakly correlated with lag time.” The figure legend now states, “Plots E and F show that threshold growth rate has a weak negative correlation with time lag, but such a correlation is not significant when using exponential growth rates.” By collecting and including more carrying capacity data, we have realized that these measurements are more variable than measurements of time lag or growth rate. Thus we do not include carrying capacity statistics, nor do we make any claims about a correlation between carrying capacity and growth rate.

- Why do you think that in defined medium you get better predictions?

That is a good question and we are not sure about it. It is interesting that the qualitative predictions do equally well, while the quantitative predictions are better in the defined medium. This could be because the complex medium is undefined, and different batches can produce slightly different results. In fact, we found that the carrying capacity of one species varied by a factor of two in different batches of the complex medium. We expect that such single-species variation will propagate to coculture experiments.

- Line 155: "at which it could have survived in the absence of a competitor". Where is that data? Why do we not see monoculture dilution rates to compare?

As mentioned above, in monoculture, *Pv* went extinct at the highest two dilution factors, while *Pa* went extinct at the highest factor. The other three species survived all dilution factors. We have added this to Fig. 4b.

- Can the model predict the ratio at which two species will co-exist? How well does it do?
- If you have an explicit model, why not fit it to the case that doesn't work with LV (lines 174ff) and see whether it does better?

Our pairwise assembly rules predict coexisting fractions of three or more species, as described in the Methods section (lines 463-469):

"For example, in a three-species coexisting community, the fraction of species 1 depends on its coexisting fractions with the other two species in pairs:

$$f_1 = (f_{12}^{w_2} f_{13}^{w_3})^{\frac{1}{w_2+w_3}}$$

where f_{12} is the fraction of species 1 after reaching equilibrium in competition with species 2, $w_2 = \sqrt{f_{21}f_{23}}$ and $w_3 = \sqrt{f_{31}f_{32}}$. Finally, these predictions are normalized by setting $f_1^* = \frac{f_1}{f_1+f_2+f_3}$."

While these quantitative predictions are good overall (see Supplementary Figure 2), the prediction that all three species in a trio will coexist is accurate only 25% of the time in the defined medium, as discussed below.

We also tried fitting our data to the LV model with added death. We used time series data from the lowest dilution factor to predict whether a pair would become coexisting or bistable at higher dilution factor(s). This qualitative prediction was successful in the case of the coexisting pair (*Ea-Pv*)—the pair was predicted to cross the coexisting region at higher dilution factors, but not for the bistable pair (*Pci-Pv*)—this pair was also predicted to cross the coexisting region at higher dilution factors. In the case of the coexisting pair, the quantitative predictions for the coexisting fraction at higher dilution factors were remarkably good: the fast grower (*Ea*) was predicted to coexist at a fraction of 0.19 at dilution factor 100 (in reality this fraction was 0.29) and at a fraction of 0.75 at dilution factor 1,000 (in reality this fraction was 0.79). Since the qualitative prediction for the bistable pair was incorrect, however, we did not pursue model fitting in the paper.

We did not try fitting the result of bistability between coexistence and exclusion, which are not allowed by LV model, to the model in Vet et. al. The failure in fitting to the simple model did not encourage us to fit to a more complicated model. Furthermore, the Vet et. al. model requires fitting 6 parameters instead of 2, so overfitting is a possibility. Overall, we believe the simple qualitative predictions of the LV model are a useful guide for our experiments, and that fitting data to a (simple or complicated) model does not give us more information.

- Why is it so hard to get coexistence of all three or four? Do you think it has something to do with medium complexity? Do you see a different outcome in the complex medium? That might be interesting to highlight.

We do observe that it is easier to get more than two species to coexist in the complex medium than in the defined medium. In trios, when the assembly rules predict coexistence of all three species, we see this occur in defined medium ~25% of the time, while in the complex medium it occurs ~75% of the time. In a prior study (Friedman et. al., 2017), predicted coexistence of all three species in trios occurred ~50% of the time. We agree that this is worth mentioning, and we have added it to the legend of Supplementary Figure 9:

"In this study, coexistence of all three species in a trio occurred ~25% of the time it was predicted in the defined medium. In the complex medium, three-species coexistence predictions were more accurate, with a success rate of ~75%."

Reviewers' Comments:

Reviewer #1:

Remarks to the Author:

The authors have done an excellent job addressing reviewer concerns. I am satisfied.

Reviewer #2:

Remarks to the Author:

The authors have addressed my concerns.

Reviewer #3:

Remarks to the Author:

I would first like to thank the authors for taking the time to consider my suggestions. I think the paper has indeed improved in its clarity and in contribution. I do have three additional comments that I think are still worth considering:

1. The 5 species predictions that are now in fig. S9 are not that impressive. I think you should add that data to Fig. S2 and mention in the main text that as number of species increases, predictions become worse. I would also be more measured in statements throughout the paper claiming that two-species dynamics predict multispecies communities. Instead, I would add: although prediction errors do increase with increasing community size.
2. A related point: Would it be possible to benchmark the prediction errors that you are measuring by calculating the error in a randomly generated dataset? Otherwise it's hard to tell whether for the 5 species an error of 0.31 is large or not.
3. A small detail: In Fig. S8 and S9, I find it strange to call them replicates if they each have different starting conditions. Instead, the biological replicates are the experiments, and you have 4 treatments in each experiment, with no technical replicates.

We thank the reviewers for their prompt responses, and Reviewer #3 for the final suggestions. We believe that these changes, explained below, help to clarify the manuscript's findings.

Reviewer #3 (Remarks to the Author):

I would first like to thank the authors for taking the time to consider my suggestions. I think the paper has indeed improved in its clarity and in contribution. I do have three additional comments that I think are still worth considering:

1. The 5 species predictions that are now in fig. S9 are not that impressive. I think you should add that data to Fig. S2 and mention in the main text that as number of species increases, predictions become worse. I would also be more measured in statements throughout the paper claiming that two-species dynamics predict multispecies communities. Instead, I would add: although prediction errors do increase with increasing community size.

We agree that the predictions for the five-species experiments are not impressive. We chose not to highlight them in the main text because there is only one five-species community, as opposed to five trios and three quads. Ideally, we would like to have data from other five-species communities before ruling out the effectiveness of the assembly rules in the case of the five-species communities. We have incorporated the results into a Supplementary Table, though, which shows all types of predictions for three, four and five species in the defined medium:

Prediction Type	Mean error / max error, 3-species communities	Mean error / max error, 4-species communities	Mean error / max error, 5-species community
Pairwise Outcomes	0.09 (0.02)	0.19 (0.02)	0.31 (0.05)
Modified Monocultures	0.30 (0.03)	0.35 (0.04)	0.30 (0.08)
Monocultures	0.40 (0.03)	0.43 (0.03)	0.36 (0.08)
Random Predictions	0.46 (0.03)	0.47 (0.03)	0.43 (0.06)

Supplementary Table 1: Accuracy of assembly rules decreases with number of species.

An extension of Table 1 from the main text shows all prediction data for three-, four- and five-species experiments in the defined medium. While the assembly rules are the best predictor of three- and four-species states, they do not offer improved accuracy in the case of the five-species community. Mean errors of three types of predictions are shown, as well as mean error for random predictions, for comparison. Monoculture predictions use carrying capacities (Fig. S5-C,D) and modified monoculture predictions also incorporate dilution and growth (in the logistic model with added death, the carrying capacity is multiplied by a factor, $(1 - \frac{\delta}{r})$). Pairwise predictions are based on the results shown in Fig. S1. Errors of quantitative predictions are the L2 norm of the distance between predicted fixed point and observed fixed point (see Methods and Supplementary Figure 2). The values shown are mean error normalized by the maximum error. Errors, shown in parentheses, are SEM of replicates (n=118 for 3-species communities, n=96 for 4-species communities; n=26 for the 5-species community; outcomes of different starting fractions of the same biological replicate were averaged before measuring error, and in the case of bistability, the smaller of the two errors was chosen).

Additionally, we have taken your suggestion to qualify the statement about multispecies predictions by emphasizing that the predictions work for **simple** multispecies communities, for example in the abstract:

tradeoff between growth and competitive ability is prevalent at low densities, causing outcomes to shift dramatically as dilution increases, and that these two-species shifts propagate to **simple** multispecies communities. Our results argue that a bottom-up approach can provide insight into how communities change under stress.

And we refer to Supplementary Table 1 in mentioning the five-species community prediction failure in the Results section:

result of five-species coculture). Overall, a quantitative generalization of our assembly rules (see Methods) predicted the equilibrium fractions with an error of 14%, significantly better than the 41% error that results from predictions obtained from monoculture carrying capacity (Table 1, Supplementary Figure 2).

Assembly rule prediction error does increase with increasing community size, however, particularly in the case of the five-species community (Supplementary Table 1, Supplementary Figure 9), which may be due to slow equilibration or infrequent coexistence of more than two species. These results indicate that pairwise outcomes are good predictors of **simple** multispecies states in the presence of increased mortality.

and in the Discussion section:

³⁶. Our results provide support for a bottom-up approach to simple multispecies communities, and show that pairwise interactions alone can generate multispecies states that appear nontrivial. Prediction errors do increase with increasing community size, however, as can be seen in the case of the five-species community (Supplementary Table 1, Supplementary Figure 9).

The aforementioned tradeoff made for striking transitions in the

2. A related point: Would it be possible to benchmark the prediction errors that you are measuring by calculating the error in a randomly generated dataset? Otherwise it's hard to tell whether for the 5 species an error of 0.31 is large or not.

We agree that random predictions are a useful benchmark. We have added them to Supplementary Table 1, as shown above, as well as to Table 1 in the main text:

Prediction Type	Mean error / max error, Trios	Mean error / max error, Quads	Overall
Pairwise Outcomes	0.09 (0.02)	0.19 (0.02)	0.14 (0.01)
Carrying Capacities (Monocultures)	0.40 (0.03)	0.43 (0.03)	0.41 (0.02)
Random Predictions	0.46 (0.03)	0.47 (0.03)	0.47 (0.02)

3. A small detail: In Fig. S8 and S9, I find it strange to call them replicates if they each have different starting conditions. Instead, the biological replicates are the experiments, and you have 4 treatments in each experiment, with no technical replicates.

We find that for many of our experiments the different starting fractions lead to the same equilibrium fraction, in which case we often think of these as (technical) replicates. We are hesitant to use the word treatment as we feel that this may imply that we expect a different final state to emerge in the different “treatment” conditions. Of course, in Supplementary Figure 8 at dilution factor 10^3 there is bistability, in which case the final fraction does depend upon the initial conditions.

To clarify these issues, we now label Day 0 as “initial fractions” and Day 7 as “final fractions,” and along the x-axis differentiate between the experiments performed in different months (which is the true variation and the primary point of Supplementary Figure 8):

Similar changes have been made to Supplementary Figure 9.